# Customer-Based Brand Equity for a Tourism Destination: The Case of Croatia

**Lenka Cervova \*** and **Jitka Vavrova**

Faculty of Economics, Technical University of Liberec, 460 01 Liberec, Czech Republic; nemeckova.j@gmail.com
\* Correspondence: lenka.cervova@tul.cz

**Abstract:** Tourism has been negatively impacted by the global COVID-19 pandemic, making it even more important for tourist destinations to focus on their brand equity from the perspective of their customers—visitors. The aim of this paper is therefore to verify and modify the model of customer-based brand equity for a tourism destination (CBBETD) and its attributes for the destination of Croatia from the perspective of Czech tourists, among whom primary research was conducted using the CAWI method (n = 451). The main CBBE dimensions were extracted using factor analysis and a model with four dimensions (awareness, image, quality and loyalty) was created. The identified attributes explain between 55% and 82% of the variability of a given dimension. Although the study's results follow the published models of CBBETD, the attributes in each dimension and the subdimension in the image dimension reflect the specificities of the destination of Croatia. Thus, the results of this paper extend the economic theory with another model and are also applicable in the field of destination management.

**Keywords:** brand equity; customer-based brand equity; Croatia; destination; destination awareness; destination brand; destination image; destination loyalty; destination management; destination quality; tourism; visitors' loyalty



## 1. Introduction

In the current era, which is heavily influenced by the global pandemic caused by COVID-19, with international tourist arrivals falling by 74% in 2020 (UNWTO 2021), it is crucial for tourist destinations to work on their brand and bear themselves in the eyes of tourists as a safe and secure place to spend their holidays. In the past decades, an increasing number of tourist destinations—cities, countries and regions—have applied marketing and branding practices to attract visitors and investors (Gertner 2011). Destination branding is one of the main topics in tourism marketing in terms of enhancing differentiation and competitiveness. This urgent need for destination branding has led to an increase in the number of investigations done on different destinations' brand equity (Oliveira and Panyik 2015).

Destination brands are very different from product brands. Destinations provide another quality than a material or financial one that can be refunded. Gartner (2014) stated, "Destinations are places of life and change". Change is the measure of brand stability, one of the main elements of branded consumer products. Destinations are multidimensional and provide different experiences to different tourists (Ruzzier 2010). Destination brands lack the brand stability that most product brands have. Several market segments consume it simultaneously; each consumer is compiling their unique product from the services on offer. Thus, destination marketers have less control over the brand experience than marketers of concrete material products or services (Hankinson 2009).

This article presents the results of a research focused on the evaluation of the CBBE destination of Croatia from the perspective of the citizens of the Czech Republic. This destination was chosen because it has been very popular in the last years in the Czech Republic. The aim was to find out what dimensions of CBBE are important in the case of

holidays in Croatia and what attributes constitute them. Croatia is the 18th most popular tourist destination in the world. Most of the tourists come from Germany, Slovenia, Austria and the Czech Republic. Tourism is one of the main sources of state revenue; it accounts for 20% of the GDP. Thanks to its location in the Mediterranean and the rugged Adriatic coast with many islands, Croatia is one of the typical summer destinations with a predominant seaside tourism. Tourists also visit many historic cities such as Dubrovnik, Split, Zadar, Sibenik or Rijeka. There are ten monuments on the UNESCO list in the country (e.g., Plitvice Lakes National Park, the historic town of Trogir or the old town of Dubrovnik) (Croatian National Tourist Board 2021). For several years, Croatia has been one of the top destinations visited by Czech residents in terms of the number of arrivals. Before the pandemic, approximately 800,000 Czech tourists visited Croatia annually (ČSÚ 2021). A sharp decline occurred in 2020, when tourism worldwide was affected by the global COVID-19 epidemic and only 481,000 Czech tourists visited Croatia, despite the fact that Croatia was one of the first countries to open its borders to Czech tourists (Ministry of Tourism 2021). It is reasonable to assume that the total number of Czech tourists in Croatia will be lower in 2021, although the destination has set favourable conditions for tourist arrivals even before the summer season. It should be mentioned that the results presented in this article were obtained by research done in 2019, when the occurrence of coronavirus infections was not anticipated.

## 2. Literature Review

### 2.1. Branding

Branding is one of the most critical tasks in the development of a marketing strategy. Kotler (1991) defined a brand as " ... a name, term, sign, symbol or design ... intended to identify the goods or services of one seller or group of sellers and to differentiate them from those of competitors". Brands are important markers of international resources and communicators of the marketing intent of an organization (Hunt 2019).

### 2.2. Brand Equity

A brand receives its value from customers by providing an image of stability, performance and other traits in reaction to a company marketing strategy. Therefore, customers know what to expect in the way of product performance. Keller (1993) named this response "customer-based brand equity". The definition of brand equity has evolved over time and academic understanding varies. Brand equity has been perceived as the added value of a product when consumers have a good impression about a brand, as the source of brand loyalty and even as the increased cash flow on branded products. Brand equity ensures higher margins compared to non-branded products. It can give a sustainable and differentiated competitive advantage (Kim and Lee 2018).

### 2.3. Customer-Based Brand Equity for a Tourism Destination

Brand equity is measured from two different perspectives. First, there is the financial value of the brand to the firm and then there is the measure of the value to the customer (Keller 2003; Pappu and Christodoulides 2017). The financial value of the brand to the firm is measured by the result of customer-based brand equity. There are several studies that developed and tested accounting methods for the appraisal of the asset value of a brand name (Lassar et al. 1995). However, our paper focuses on brand equity from the perspective of the value to the customer.

Customer-based brand equity (CBBE) is at present more than 20 years old and a well-developed construct, the roots of which lead us to the 1980s (Fayrene and Lee 2011). During these years, this concept received much attention (Ruzzier 2010). The CBBE concept was defined "as the differential effect that brand knowledge has on consumer response to the marketing of that brand" (Keller 1998). There have been numerous attempts to summarize measures of brand equity, approaching the construct from different perspectives. The Table 1 below demonstrates those dimensions (Almeyda and George 2020).

**Table 1.** Customer-based brand equity dimensions.

| Aaker (1991) | Keller (1993, 1998, 2003) | Lassar et al. (1995) | Konecnik and Gartner (2007) | San Martín et al. (2019) |
|---|---|---|---|---|
| Brand awareness | Brand salience | Performance | Destination awareness | Destination awareness |
| Brand perceived quality | Brand performance Brand imagery | Social image | Destination perceived quality | Destination quality |
| Brand association | Brand judgements Brand feelings | Price/value Trustworthiness | Destination image | Destination image Destination satisfaction |
| Brand loyalty | Brand resonance | Identification/attachment | Destination loyalty | Destination loyalty |

Source: (Almeyda and George 2020).

The basic concept of CBBE is that the measure of the brand strength depends on how consumers feel, think and act with respect to the brand. To achieve consumer resonance a brand first needs to elicit emotional reactions from consumers. To achieve that, a brand must have an appropriate identity and the right meaning. At best, customers therefore consider the product as relevant and "their kind" (Koththagoda 2017). The model of customer-based brand equity for a tourism destination was proposed and verified by Konecnik and Gartner (2007). It was confirmed that the level of CBBETD is positively related to an extent to destination brand equity dimensions, which are presented further.

### 2.4. Dimensions of the Customer-Based Brand Equity

Based on the CBBE model, Konecnik and Gartner (2007) have investigated the different dimensions of customer-based brand equity for a tourism destination (CBBETD). Our paper continues their work, which listed awareness, image, quality and loyalty as the dimensions of a destination as antecedents to CBBETD. Tourists from different backgrounds sense various dimensions of a destination distinctly.

#### 2.4.1. Destination Awareness

The term destination awareness was introduced in behavioural studies of consumer and was described in the tourism decision process by Goodall and Ashworth (1993). Aaker (1991) defined destination awareness as "the ability of a potential buyer to recognize or recall that a brand is a member of a certain product category". Brand awareness increases a destination's potential of being preferred more often than other unknown destinations (Kladou and Kehagias 2014). It also brings a better chance of being chosen by potential customers among all rival destination brands (Hoyer and Brown 1990).

Staying focused on destination brand awareness is important because it provides optimistic information and creates positive emotions that are likely to increase the possibility of making a purchase (Baldauf et al. 2003). Destination brand awareness also plays a critical role in tourists' destination quality perception (Buil et al. 2013; Nikabadi et al. 2015).

Awareness is only the first and necessary step in the decision process, and may lead to visit a destination; on the other hand, it is insufficient, because the very awareness provides only a set of choice (Goodall and Ashworth 1993). For getting more tourist visits, destination brand must first achieve awareness and then a positive destination image.

#### 2.4.2. Destination Image

Destination image is formed by the interaction of people and places (Pearce and Stringer 1991). Based on subjective interpretations, a tourist's thoughts and feelings toward the destination are generated and affect their image formation (Tasci et al. 2007; Veasna et al. 2013). Destination image is described as "the sum of beliefs and impressions that a person has of a destination" (Chiu et al. 2014).

Despite the significant effect of destination image on CBBETD, only a limited amount of research has focused on the moderating effect of destination image. Line and Hanks (2016) identified the moderating effect of destination image in relation to guests' perceptions and

behavioural intentions in the green hotel industry. Other researchers have considered destination image as an antecedent of the intention to revisit a destination (Stylos et al. 2016) or as an outcome of destination marketing (Wong et al. 2016).

For the purpose of this paper, destination image represents "an interactive system of thoughts, opinions, feelings, visualizations, and intentions toward a destination" (Tasci et al. 2007). It has been proven that destination image has a large impact on customer loyalty. The image of a destination is the most important and significant dimension of CBBETD model. A leading destination image brings more customers to make an effort to visit or revisit a destination and also to recommend it (behavioural and attitudinal loyalty). Destination image creates an impact on loyalty through satisfaction (Marine-Roig 2021).

### 2.4.3. Destination Quality

Another key aspect of CBBETD is the quality of a destination. Destination quality is defined as a visitor's evaluation of the standard of tourism products at the destination (infrastructure of attractions, tourist facilities and services). Tourists judge if the destination products meet their requirements or expectations according to their real perceptions (Le Chi 2016).

Nevertheless, quality measurement is a very difficult and complex process. In order to find out the quality, it is necessary to research the tourists' evaluation of products and services and the tourists' experience in the destination. All these elements affect consumer behaviour and preference. The aspect of destination quality is the most important component of CBBETD. When researching destination quality, attention should be paid to a distinction between perceived quality and tourists' satisfaction (Ruzzier 2010).

### 2.4.4. Destination Loyalty

From a marketing perspective, loyalty is defined as customers' behaviour or intentions to re-buy or re-patronize certain product or service, causing repetitive purchasing of the same brand products (Hawkins et al. 1995). Loyalty measures a consumer's strength of affection towards a brand. It is based on a consumer brand preference or their intention to buy a product of a certain brand. Customer satisfaction, customer experience, value, service quality, performance, price and brand name all contribute to loyalty (Backman and Crompton 1991). In destination brand research loyalty plays a big role, but it should be examined in a long-term range. It can serve as a useful tool for prediction of future destination choice (Oppermann 2000).

### 2.5. Executed Research on Customer-Based Brand Equity

The concept of CBBETD started to be tested for many destinations by various researchers and from many perspectives more than 10 years ago. For example, Boo et al. (2009) measured the CBBE for Las Vegas and Atlantic City. However, in contrast to our paper, besides awareness, image quality and loyalty, they added another dimension of destination brand value to their model. Yousaf and Amin (2017) measured the CBBE for a tourist destination named the Kashmir valley in India. Their study suggests particular steps to ensure a strong brand equity of the Kashmir valley. Almeyda and George (2020) compared the CBBEs of Puerto Rico and the US Virgin Islands while using different dimensions of the CBBETD model (value, social image, performance, trustworthiness and identification). Their study claims that the core dimension that explains more than ninety percent of the customer-based brand equity is brand performance, which is a substitute of the destination quality in our CBBETD model.

The study executed by Suta et al. (2019) investigated empirical information for testing the concept of cultural differences on the integration of variables in the CBBETD. The subject of their research was the tourist destination of Bali. Furthermore, their research applied the CBBETD model to investigate cultural differences as a mediating indicator of the correlation among brand loyalty and other indicators in the CBBETD.

Another study that needs to be listed is an empirical CBBETD study of the Liberec region in the Czech Republic executed by the authors of this paper (Červová and Pavlů 2018). The previous study used the same dimensions of CBBETD and also tested the concept very well.

Based on the literature review, the following research questions are addressed in this study:

– RQ 1: Is the model of CBBETD proposed by Ruzzier (2010) applicable also to Croatia from the perspective of Czech visitors?
– RQ 2: Are the dimensions of the proposed model identical?
– RQ 3: Are there any subdimensions that can be identified?

### 3. Methodology

The purpose of this paper is to verify and modify the CBBETD model in the context of the destination of Croatia from the perspective of Czech visitors. The research methodology is based on the CBBETD concept introduced and modified by Ruzzier (2010). This concept of brand equity consists of four subdimensions, namely, awareness, image, quality and loyalty. Since the attributes within the subdimensions of awareness (three attributes) and loyalty (three attributes) are generally applicable regardless of the destination, they were adopted without change from the original model by Ruzzier (2010). However, the attributes included in the image and quality subdimensions had to be adapted to fit the characteristics of the destination. To this end, focus group research was conducted in the first phase of the research, involving 25 potential respondents. The aim of the focus group interviews was to identify suitable attributes specific to Croatia from the perspective of Czech visitors that would have an impact on image and quality. The focus groups were conducted with an emphasis on subjective perceptions, expectations and experiences; therefore, no attribute options were presented to the participants in order not to influence their opinion. The output of the focus groups was 21 attributes falling into the image subdimension and 9 attributes falling into the quality subdimension. Thus, the brand equity of the destination of Croatia was measured using a total of 36 attributes.

In the second phase of our research, the data were collected through a structured questionnaire (see the Appendix A) using the method of online interviewing (CAWI). The first part of the questionnaire consisted of questions characterizing the respondents' travels to Croatia (such as frequency of visits, length of stay, sources of information, way of organizing holidays, etc.). The second part of the questionnaire focused on the attributes of CBBETD, which were transformed into statements and rated on a scale of 1 to 5, with 1 indicating total disagreement and 5 total agreement with the statement.

The respondents were selected by a quota selection method according to gender and age so that the sample would correspond to the profile of a Czech visitor to Croatia (Czechtourism 2019). However, only people over 18 years of age could participate in the survey. The data were analysed using IBM SPSS Statistics software. Factor analysis using principal components and the varimax rotation method was performed to identify significant attributes determining the four subdimensions of CBBETD. The appropriateness of using exploratory factor analysis was verified using Barlett's test of sphericity, which showed significant correlations in the correlation matrix (value of 0.000 for all analyses performed). The validity and reliability for each of the subdimensions were verified using Kaiser–Meyer–Olkin (KMO) values and Cronbach's alpha coefficients. All variables could be considered valid as KMO values ranged from 0.701 to 0.940. The values of Cronbach's alpha coefficients were 0.803 to 0.934, indicating acceptable to excellent results. The identified factors within the four subdimensions explained 55.11 to 81.03% of the total variability. Three factors were identified in the image subdimension and one factor each in the other subdimensions (see Table 2).

**Table 2.** Validity and reliability check.

| Dimension | Number of Attributes Assigned to Subdimension | KMO | Cronb. Alpha | Total Variance Explained (%) | Number of Extracted Factors (Attributes) |
|---|---|---|---|---|---|
| Awareness | 3 | 0.701 | 0.803 | 72.18 | 1 (3) |
| Image | 21 | 0.940 | 0.934 | 60.32 | 3 (20) |
| Quality | 9 | 0.888 | 0.873 | 55.11 | 1 (8) |
| Loyalty | 3 | 0.706 | 0.879 | 81.03 | 1 (3) |

Source: own processing.

There were 465 completed questionnaires. Nevertheless, the elimination of problematic questionnaires reduced the sample size to 451. In terms of gender of the visitors, 47% were male and 53% female. Out of the total number of respondents, 30% of tourists were 18–30-year-olds, 22% were 31–40-year-olds, 25% of respondents were 41–50-year-olds, 12% were 51–60-year-olds and 11% were older than 61. As per the monthly net income of the household it was found that 31% earned less than CZK 25,000, 22% earned in the range of CZK 25,001–35,000, 19% in the range of CZK 35,001–45,000, 11% in the range of CZK 45,001–55,000 and 17% earned more than CZK 55,000 a month (Table 3).

**Table 3.** Sample characteristics.

| Number of Respondents | N | 451 |
|---|---|---|
| Sex | male | 46.80 |
| | female | 53.20 |
| Age | 18–30 | 29.70 |
| | 31–40 | 22.40 |
| | 41–50 | 25.10 |
| | 51–60 | 11.80 |
| | 61 and older | 11.10 |
| Income (CZK) * | less than 25,000 | 31.10 |
| | 25,001–35,000 | 21.80 |
| | 35,001–45,000 | 19.00 |
| | 45,001–55,000 | 11.30 |
| | 55,001 and more | 16.80 |

* Exchange rate (3 November 2021): 25.50 CZK/1 EUR. Source: own processing.

## 4. Results

A factor analysis was conducted to test and eliminate attributes within the four CBBETD subdimensions. The first subdimension examined was awareness. As can be inferred from Table 4, respondents rated awareness very well (means ranging from 4.34 to 4.40). All three attributes examined reached a factor loading of more than 0.500, thus constituting a single factor ("awareness"), explaining 72.18% of the total variability.

**Table 4.** Awareness.

| Variables | Mean | Factor Loading |
|---|---|---|
| Popular TD | 4.40 | 0.869 |
| Attractive and known TD | 4.34 | 0.867 |
| Imagining of TD | 4.35 | 0.810 |
| **% Variance extracted** | **72.18** | |

Note: TD = tourist destination. Source: own processing.

The second subdimension analysed was image. In this case, the factor analysis was conducted a total of three times with the successive elimination of variables that were not

part of either factor. The aim of this procedure was to eliminate variables with low factor loading (less than 0.500) and to explain as much of the variability as possible. The third factor analysis identified three factors explaining 60.32% of the variability (see Table 5). The first factor, named attractions, includes variables such as towns and villages, nature, cultural attractions, beaches, mountains and historical attractions. The second factor can be named amenities and includes opportunities for water recreation, opportunities for recreational activities, wide range of gastronomy and accommodation facilities, pleasant weather, summer destination, friendly and hospitable people and easy accessibility. Within the image subdimension, a third factor was also identified and named ambiance. It contains variables such as modern wellness resorts, shopping facilities, exciting atmosphere, good nightlife and entertainment. Looking closely at the averages of all variables within the image subdimension, it is clear that the variables that respondents rated the highest were summer destination, opportunities for recreational activities including water recreation, pleasant weather, relaxing atmosphere (means from 4.18 to 4.63). On the other hand, the lowest rated variables were wellness resorts (2.98), shopping facilities (3.30) and exciting atmosphere (3.45). Similar results emerged from the qualitative study (focus groups) in which participants most frequently mentioned Croatia as a summer, relaxing destination with many opportunities for recreation at the seaside, including a variety of beaches.

**Table 5.** Image.

| Variables | Mean | Factor Loading | | |
| --- | --- | --- | --- | --- |
| | | Attractions | Amenities | Ambiance |
| Lovely towns and villages | 4.04 | 0.705 | 0.339 | 0.239 |
| Beautiful nature | 4.16 | 0.692 | 0.392 | 0.157 |
| Interesting cultural attractions | 3.75 | 0.687 | 0.197 | 0.437 |
| Beautiful beaches | 4.00 | 0.684 | 0.339 | 0.113 |
| Beautiful mountains | 3.91 | 0.678 | 0.180 | 0.163 |
| Interesting historical attractions | 3.78 | 0.658 | 0.144 | 0.462 |
| Good opportunities for water recreation | 4.27 | 0.152 | 0.742 | 0.254 |
| Good opportunities for recreation activities | 4.34 | 0.278 | 0.738 | 0.155 |
| Pleasant weather | 4.22 | 0.317 | 0.717 | 0.076 |
| Wide range of gastronomy facilities, local food | 4.00 | 0.171 | 0.702 | 0.391 |
| Summer destination | 4.63 | 0.122 | 0.672 | −0.133 |
| Wide range of accommodation facilities | 4.11 | 0.207 | 0.670 | 0.316 |
| Friendly and hospitable people | 4.07 | 0.267 | 0.660 | 0.206 |
| Transportation accessibility | 4.10 | 0.235 | 0.646 | 0.125 |
| Relaxing atmosphere | 4.18 | 0.438 | 0.614 | 0.117 |
| Good opportunities for adventure | 3.92 | 0.197 | 0.545 | 0.534 |
| Modern wellness resorts | 2.98 | 0.189 | −0.017 | 0.788 |
| Good shopping facilities | 3.30 | 0.139 | 0.143 | 0.758 |
| Exciting atmosphere | 3.45 | 0.290 | 0.222 | 0.682 |
| Good nightlife and entertainment | 3.84 | 0.297 | 0.348 | 0.526 |
| **% Variance extracted** | | **60.32** | | |

Source: own processing.

Within the third subdimension "quality", one factor explaining 55.11% of the variability (see Table 6) was identified. A factor analysis was conducted twice in total, with the successive elimination of variables that did not reach a factor loading of 0.500. The quality subdimension included variables such as quality of gastronomy, services, accommodation, infrastructure, unpolluted environment, good value for money and personal safety. The latter two variables were also rated the highest by respondents—a mean of 3.85 for personal

safety and a mean of 3.82 for good value for money. In contrast, the lowest rated attribute was the level of cleanliness (3.30).

**Table 6.** Perceived quality.

| Variables | Mean | Factor Loading |
|---|---|---|
| High quality of gastronomy | 3.54 | 0.839 |
| High quality of services | 3.56 | 0.821 |
| High level of cleanliness | 3.30 | 0.774 |
| Unpolluted environment | 3.63 | 0.746 |
| High quality of accommodation | 3.61 | 0.728 |
| High quality of infrastructure | 3.37 | 0.722 |
| Good value for money | 3.82 | 0.645 |
| High level of personal safety | 3.85 | 0.641 |
| **% Variance extracted** | **55.11** | |

Source: own processing.

The last subdimension was loyalty. The factor analysis performed showed high loadings on a single factor ("loyalty"), which explained 81.03% of the total variability (see Table 7). Tourists visiting Croatia would recommend a visit to this destination to their friends and acquaintances (mean of 3.90) and would also visit again in the future (mean 3.86). The evaluation of the variable choice of Croatia as a holiday destination was slightly worse, even if the cost of a holiday in Croatia increased (mean of 3.24).

**Table 7.** Loyalty.

| Variables | Mean | Factor Loading |
|---|---|---|
| Recommend TD | 3.90 | 0.929 |
| Visit TD in future | 3.86 | 0.924 |
| Visit TD even if costs increase | 3.24 | 0.845 |
| **% Variance extracted** | **81.03** | |

Note: TD = tourist destination. Source: own processing.

## 5. Conclusions

In conclusion, it can be argued that the model of customer-based brand equity for a tourism destination proposed by Ruzzier (2010) can be used in a modified form (Figure 1) for the destination of Croatia from the perspective of Czech tourists (referring to RQ 1). In respect of the RQ 2 it can be stated that our modified concept of CBBETD consists of the same dimensions of awareness, image, quality and loyalty.

The model is very useful as it provides Croatia with strategic options to improve its position in the eyes of current and potential tourists. However, the outputs of the factor analyses showed that the only problematic item compared to the original model was the image subdimension, the attributes of which were already modified on the basis of qualitative research in the form of focus groups to reflect the specifics of the destination. Regarding RQ 3, the results of this research showed that the image subdimension is made up of three factors, namely attractions, amenities and ambiance. Thus, the results of this research build on the work of Cooper et al. (2005) who identified six "As" of a destination: attractions, accessibility, amenities, ancillary services, available packages and activities. The variables under the latter mentioned factor of ambience (modern wellness resorts, good shopping facilities, exciting atmosphere, good nightlife and entertainment) scored worse compared to the other variables of the image subdimension, indicating room for improvement in order to increase the brand equity of this destination. Other variables that Croatia as a destination should focus on improving were cleanliness and quality of infrastructure (quality subdimension). Furthermore, it was found that although

respondents expressed high levels of loyalty, there was a lower willingness to visit Croatia even if costs were to increase. This is something that Croatia should be careful about and the increase in costs or prices should be accompanied by an increase in the value offered, so that tourists do not prefer other competitive destinations. A destination brand is a competitive identity that distinguishes a place from others. The destination brand should be the basis for the strategy of all destination management organisations as well as for communication with the public. Thus, the Croatian National Tourist Board can use the results of this research in its concept.

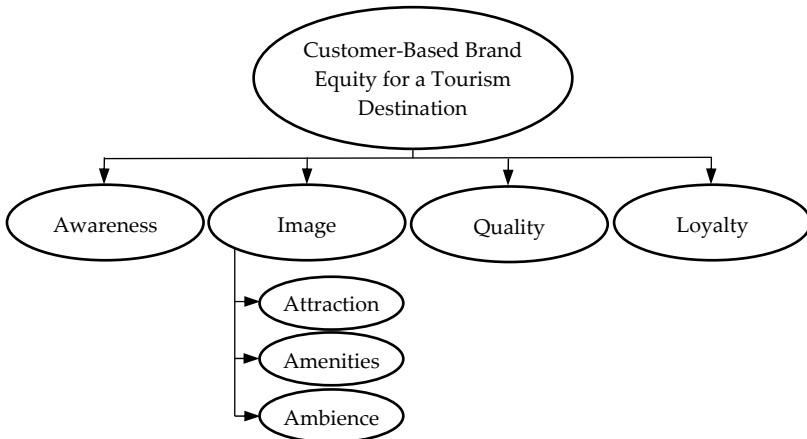

**Figure 1.** Customer-based brand equity model for Croatia. Source: own processing.

As noted above, data collection was conducted in the pre-COVID-19 period, so further research should look at how COVID-19 has affected customer-based brand equity. It is likely that tourists' perspectives on different destinations may have changed due to new circumstances, and a new dimension could have been added to the CBBETD model as well, to assess the set conditions for travel in the COVID-19 pandemic. It should also be noted that this model is based on visitor opinion only. A more holistic view of the brand equity of a tourist destination could be provided by a deeper analysis that also takes into account the views of other stakeholders (residents, local businesses and their employees). Furthermore, future research could focus on the relationship of individual factors (attributes) to overall satisfaction, for example by integrating Kano's model (Kano 1984), which would identify those factors (attributes) that contribute most to increasing visitor satisfaction and therefore to increasing visitor loyalty.

**Author Contributions:** Conceptualization, L.C. and J.V.; methodology, L.C.; formal analysis, J.V.; investigation, L.C.; resources, J.V.; data curation, L.C.; writing—original draft preparation, L.C. and J.V.; writing—review and editing, L.C. and J.V.; visualization, L.C. and J.V.; All authors have read and agreed to the published version of the manuscript.

**Funding:** This research received no external funding.

**Institutional Review Board Statement:** Not applicable.

**Informed Consent Statement:** Not applicable.

**Data Availability Statement:** Not applicable.

**Conflicts of Interest:** The authors declare no conflict of interest.

## Appendix A. Questionnaire

1. How many times have you visited Croatia in last 3 years?
   (1) 1x        (2) 2x        (3) 3x        (4) 4x and more
   (5) I haven't been to Croatia in last 3 years. => *Continue to Q. 8*
2. How long did your last stay in Croatia take?
   (1) 3–6 days            (3) 11–14 days
   (2) 7–10 days           (4) More than 15 days
3. Who did you spend your last stay in Croatia with?
   (1) I was there alone.              (3) Partner              (5) Work colleagues
   (2) Friends                         (4) Family               (6) Other, please specify:
4. How did you organize your last stay in Croatia?

   |  | by myself | travel agency/tour operator |
   |---|:---:|:---:|
   | Transportation | ☐ | ☐ |
   | Accommodation | ☐ | ☐ |
   | Meals | ☐ | ☐ |
   | Programme | ☐ | ☐ |

5. How did you get to Croatia to spend your last holiday?
   (1) car            (3) airplane        (5) combination
   (2) coach bus      (4) train           (6) other, please specify:
6. Where were you accommodated during your last stay in Croatia?
   (1) Hotel          (3) Apartment                      (5) Bed and breakfast
   (2) Campsite       (4) My friends' or relatives' house (6) Others, please specify:
7. What type of meal plan did you choose for your last stay?
   (1) Self-catering            (3) Breakfast and dinner included    (5) All inclusive
   (2) Breakfast included       (4) All meals included
8. How do you perceive Croatia as a tourist destination? For each statement, please choose if you strongly disagree, disagree, neither agree nor disagree, agree or strongly agree.

| | Strongly disagree | Disagree | Neither agree nor disagree | Agree | Strongly agree |
|---|:---:|:---:|:---:|:---:|:---:|
| Croatia is a popular tourist destination. | ☐ | ☐ | ☐ | ☐ | ☐ |
| I can easily imagine how the holidays in Croatia look like. | ☐ | ☐ | ☐ | ☐ | ☐ |
| Croatia is quite attractive and known. | ☐ | ☐ | ☐ | ☐ | ☐ |
| Croatia has a beautiful nature. | ☐ | ☐ | ☐ | ☐ | ☐ |
| Croatia has beautiful mountains. | ☐ | ☐ | ☐ | ☐ | ☐ |
| Croatia has beautiful beaches. | ☐ | ☐ | ☐ | ☐ | ☐ |
| Croatia has lovely towns and cities. | ☐ | ☐ | ☐ | ☐ | ☐ |
| Croatia has attractive cultural attractions. | ☐ | ☐ | ☐ | ☐ | ☐ |
| Croatia has interesting historical attractions. | ☐ | ☐ | ☐ | ☐ | ☐ |
| Croatia offers good opportunities for nightlife and entertainment. | ☐ | ☐ | ☐ | ☐ | ☐ |
| Croatia offers good opportunities for recreation activities. | ☐ | ☐ | ☐ | ☐ | ☐ |
| The people in Croatia are friendly and hospitable. | ☐ | ☐ | ☐ | ☐ | ☐ |
| Croatia has a pleasant weather. | ☐ | ☐ | ☐ | ☐ | ☐ |
| Croatia is politically stable. | ☐ | ☐ | ☐ | ☐ | ☐ |
| Croatia has a wide range of accommodation facilities. | ☐ | ☐ | ☐ | ☐ | ☐ |
| Croatia has good opportunities for water recreation. | ☐ | ☐ | ☐ | ☐ | ☐ |
| Croatia has a wide range of gastronomy facilities and offers local food. | ☐ | ☐ | ☐ | ☐ | ☐ |
| Croatia offers good opportunities for adventure. | ☐ | ☐ | ☐ | ☐ | ☐ |
| Croatia is easily accessible regarding transportation. | ☐ | ☐ | ☐ | ☐ | ☐ |
| Croatia has a relaxing atmosphere. | ☐ | ☐ | ☐ | ☐ | ☐ |
| Croatia is a summer destination. | ☐ | ☐ | ☐ | ☐ | ☐ |
| Croatia offers modern wellness resorts. | ☐ | ☐ | ☐ | ☐ | ☐ |
| Croatia has good shopping facilities. | ☐ | ☐ | ☐ | ☐ | ☐ |
| I can easily speak Czech in Croatia. | ☐ | ☐ | ☐ | ☐ | ☐ |
| Croatia has exciting atmosphere. | ☐ | ☐ | ☐ | ☐ | ☐ |
| Croatia is safe and secure. | ☐ | ☐ | ☐ | ☐ | ☐ |
| Croatia has a high quality of accommodation | ☐ | ☐ | ☐ | ☐ | ☐ |
| Croatia has a high quality of infrastructure. | ☐ | ☐ | ☐ | ☐ | ☐ |

| | | | | | |
|---|---|---|---|---|---|
| Croatia has a high level of cleanliness. | ☐ | ☐ | ☐ | ☐ | ☐ |
| Croatia has a high quality of gastronomy services. | ☐ | ☐ | ☐ | ☐ | ☐ |
| Croatia has a high quality of services. | ☐ | ☐ | ☐ | ☐ | ☐ |
| Croatia offers good value for money. | ☐ | ☐ | ☐ | ☐ | ☐ |
| Croatia has an unpolluted environment. | ☐ | ☐ | ☐ | ☐ | ☐ |
| I intent to visit Croatia again. | ☐ | ☐ | ☐ | ☐ | ☐ |
| I would like to recommend Croatia to my friends and relatives. | ☐ | ☐ | ☐ | ☐ | ☐ |
| I would choose Croatia for my holiday even if the costs were higher. | ☐ | ☐ | ☐ | ☐ | ☐ |

9. Gender:　(1) Male　　　　　(2) Female

10. Age:　(1) 18–30　　　　　(3) 41–50　　　　(5) 61 and older
　　　　　(2) 31–40　　　　　(4) 51–60

11. Net monthly household income (CZK)　(1) Less than 25,000
　　　　　　　　　　　　　　　　　　　　(2) 25,001–35,000
　　　　　　　　　　　　　　　　　　　　(3) 35,001–45,000
　　　　　　　　　　　　　　　　　　　　(4) 45,001–55,000
　　　　　　　　　　　　　　　　　　　　(5) 55,001 and more

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
