# Peer review of "Customer-Based Brand Equity for a Tourism Destination: The Case of Croatia"

_economies, doi:10.3390/economies9040178_

Round 1
Reviewer 1 Report
The article is part of the contemporary research trend in economics.
The theme of the brand of tourist destinations is often discussed in scientific publications. However, this is not a disadvantage of this article. Despite many publications on the subject, tourist destination managers cannot correctly shape the brand of their regions.
Therefore, the scientific article submitted for review should be considered up-to-date and necessary. It should reach a wide range of readers.
The methodology of the conducted research is correct and appropriate for this type of economic subject.
The article reviews considerable scientific achievements regarding the brand of tourist destinations.
Forty-two scientific publications were used, but as many as 26 of them are older than those from 2015.
Citing their own publication in verse 174, the authors revealed their names. However, this does not conflict with the reviewer. However, the article cited in this way should be hidden from the review, and only disclosed when publishing the article.
Tables and figures are correct and legible.
The article has several disadvantages that need to be removed:
1. The authors did not make a clear research hypothesis. Thus, it was not verified as a result of the conducted survey research.
2. The research aims in the abstract and in verse 189 are different. They should be standardized.
3. The authors did not justify why they chose the CBBETD model, the Ruzzier and Gartner (2007) and not the other. Is this the best one? Does it best suit the research topic? The justification for this choice should be provided.
4. In verse 207 the authors refer to the profile of a Czech visitor to Croatia. They do not state who developed such a profile. The source must be provided.
5. What method was used to calculate the size of the research sample of respondents to the survey? (verse 226) Provide a formula.
6. To better understand the study, Table 3 should show the exchange rate of the Czech koruna (CZK) exchange rate to the US dollar or the euro and the average national wage.
7. San Martin, Herrero I García (2018), quoted in Table 3, does not have in a reference list.
The authors note that they conducted their research before the COVID-19 pandemic. This is a fundamental limitation of the test results and their interpretation. However, this does not diminish the importance and significance of this study. It may also inspire to repeat this research in the changed economic reality soon.
In their study, the authors include personal safety as a quality variable. It should be considered whether safety is such an essential factor influencing the brand of a tourist destination that it should be a separate fifth factor. Personal safety and economic security can be distinguished here (e.g. price stability, inflation rate, the price level concerning prices in the Czech Republic, etc.). This is a problem for further scientific discussion.
The article also has a poorly developed last part - conclusions. This part of the article should be more elaborate, and villages should be more specific. There should be a statement about the achievement of the research goal and verification of the research hypothesis.
Reviewer 2 Report
Customer-based brand equity for a tourism destination: The case of Croatia
The topic is interesting and the sample of respondents is considerable, but it is necessary to reinforce the theoretical background, improve readability, expand the information on the case study, and adapt the structure and format to the MDPI style (please see “Instructions for authors”).
Keywords: Three to ten pertinent keywords need to be added after the abstract. We recommend that the keywords are specific to the article, yet reasonably common within the subject discipline.
Some keywords are missing such as: destination awareness; destination image; destination quality; visitors’ loyalty
To improve readability, in the titles of sections, tables and figures, it is advisable to avoid acronyms. For example: FROM: Table 1. CBBE dimensions TO: Table 1. Customer-based brand equity dimensions.
2. Theoretical background
Between subsections 2.1 and 2.2, it is necessary to include a subsection to define a crucial concept in this research, “brand equity”, and to review the related literature. Please see “Brand equity of a tourist destination” (MDPI Sustainability, 2018).
In the current subsection 2.2, a major article “Customer-based brand equity for a destination” (ATR, 2007) is missing.
2.3.2. Destination Image. The literature review on destination image is weak. For example, the destination image definition of Chiu et al. (2014) is a faulty copy of a popular definition (Crompton, 1979). In this theoretical framework, the relationships between the image of the destination and the satisfaction and loyalty of visitors are also lacking. Please see “Measuring online destination image, satisfaction, and loyalty” (MDPI TourHosp, 2021) and related literature in this article.
At the end of section 2, you should include a subsection dedicated to the proposed conceptual model, to show a diagram with the constructs, hypotheses, and relationships between them based on the critical review of the previous literature.
Materials and Methods
FROM: 2. Methodology TO: 3. Materials and Methods
It is necessary to include a subsection to describe the case study and its context. Some of the details about Croatian tourist destination in the introduction should be here.
Results and Discussion
FROM: 3. Data Analyses and Discussion TO: 4. Results and Discussion
Appendix:
The survey questionnaire is missing, which should include the constructs and items, according to the conceptual model, as well as the references from previous research that serve as the foundation. Without a solid basis, the results seem arbitrary.
References:
Please check the citations and references: Multidisciplinary Digital Publishing Institute (MDPI) style. I use Mendeley.com with the MDPI style.
Round 2
Reviewer 2 Report
Customer-based brand equity for a tourism destination: The case of Croatia
The article has improved, but I see some minor changes necessary before publication. An empirical research should have a results section, so I recommended changing "Data analyses" to "Results". The fact of including a conceptual model and a questionnaire, well-structured and substantiated, increases the chances that the article will be cited by other researchers. It would also be interesting to include, in the Materials and Methods section, a subsection to describe the case study and its context, because I suppose there are many readers who have not had the pleasure of visiting Croatia.
References:
Sorry! I believed that the publisher had unified the (MDPI) style of the bibliography in all their publications, but I see that "economies" recommends the Chicago-MDPI style.
In any case, the reference list should not include the in-text citation in parentheses. Please, in journal titles, capitalise all major words.
